# Experiential Characteristics among Individuals with Tinnitus Seeking Online Psychological Interventions: A Cluster Analysis

**DOI:** 10.3390/brainsci12091221

**Published:** 2022-09-09

**Authors:** Eldre W. Beukes, Srikanth Chundu, Pierre Ratinaud, Gerhard Andersson, Vinaya Manchaiah

**Affiliations:** 1Vision and Hearing Sciences Research Group, School of Psychology and Sport Science, Anglia Ruskin University, Cambridge CB1 1TP, UK; 2Virtual Hearing Lab, Collaborative Initiative between University of Colorado School of Medicine and University of Pretoria, Aurora, CO 80045, USA; 3LERASS Laboratory, University of Toulouse, BP67701 Toulouse, France; 4Department of Behavioral Sciences and Learning, Department of Biomedical and Clinical Sciences, Linköping University, SE58183 Linköping, Sweden; 5Department of Clinical Neuroscience, Division of Psychiatry, Karolinska Institute, 17176 Stockholm, Sweden; 6Department of Otolaryngology–Head and Neck Surgery, University of Colorado School of Medicine, Aurora, CO 80045, USA; 7UCHealth Hearing and Balance, University of Colorado Hospital, Aurora, CO 80045, USA; 8Department of Speech-Language Pathology and Audiology, University of Pretoria, Gauteng 0002, South Africa; 9Department of Speech and Hearing, School of Allied Health Sciences, Manipal 756104, Karnataka, India

**Keywords:** tinnitus, social representations, attitude, subgroups, phenotype

## Abstract

Objective: This study was designed to gain insights regarding patterns of social representations (values, ideas, beliefs) of tinnitus and their relation to demographic and clinical factors. Method: A cross-sectional survey design was used including 399 adults seeking help and reporting interest in internet-based cognitive behavior therapy for tinnitus. Data were collected using a free association task and analysis used qualitative (content analysis) and quantitative (cluster analysis and chi-square analysis) using the Iramuteq software. Results: The social representations identified the negative impact of tinnitus and included the way it sounded (descriptions of the way tinnitus sounds (18%), annoyance (13.5%), and persistence (8%)). Four clusters were identified representing four levels of tinnitus severity, namely debilitating tinnitus (24%), distressing tinnitus (10%), annoying tinnitus (46%), and accepting tinnitus (20%). Cluster identity was associated with demographic and clinical variables. Discussion: The identified clusters represented tinnitus severity experience in four stages, ranging from debilitating tinnitus to acceptance of tinnitus. These findings are important for clinical practice where tinnitus descriptions can indicate the stage of the tinnitus experience and which intervention pathway may be most appropriate.

## 1. Introduction

Disability, physical and/or mental health difficulties result in activity limitations and participation restrictions and can negatively impact upon quality of life [1]. Addressing these difficulties is complicated as various factors may contribute such as environmental factors as highlighted in the International Classification of Functioning, Disability and Health Framework [1]. This dynamic interaction between a person’s health condition, environmental factors and personal factors will affect individual function [2]. This view is in line with a “biopsychosocial” model of disability, which acknowledges the combination of physical and contextual factors that can limit activity and restrict participation. It is also concordant with the social representations theory (SRT), which suggests that systems of values, ideas, and practices influence worldviews [3]. This encompasses social norms and social and individual attitudes towards disability, difficulties with the problem, and help-seeking [4]. The SRT deals with our beliefs about the world and social interactions with others and our knowledge about the world. It finds its place in social psychology and suggests that social groups develop a comprehensible understanding of different aspects of reality, leading to individuals perceiving the surrounding world. These models and theories suggest that health and disability are not individual entities but rather interactions between various personal factors, activities and environmental factors. Beliefs are seen as being influenced by the social world and group beliefs. These beliefs may, however, differ depending on the stage and social situations of the presenting health difficulty.

Social representations have been explored in the context of hearing loss [5,6]. These studies show that social representations of hearing loss differ in different countries, with those from India for instance having more positive associations to hearing loss compared to participants from other countries. Furthermore, the most important representations of hearing loss were assessment and management; causes of hearing loss; communication difficulties; disability; hearing ability or disability; hearing instruments; negative mental state; the attitudes of others; and sound and acoustics of the environment. A study on social representations of hearing aids [7] also noted an influence of country on perceptions regarding the use of hearing aids. Participants from the UK were more likely to associate hearing aids with cosmetics, whereas those from Iran associated hearing aids with aging and disability. Significant aspects regarding hearing aids were disability and aging; appearance and design; costs; hearing instruments; and improved hearing and communication. These studies provide new understanding of how collectively a group of individuals with different social norms think about hearing loss and hearing aids and add to the research on stigma theory which is a common theoretical approach used in hearing healthcare research.

Although tinnitus is one of the most common hearing-related difficulties, affecting at least 10% [8] of the adult population, we have limited understanding on the mechanisms behind tinnitus. Tinnitus is defined as the perception of sound in the absence of an external sound source. It is a sensation of sound in the absence of an external acoustic source; although the types of sound persons with tinnitus hear may vary and reactions to tinnitus sounds also differ substantially between individuals. For a significant proportion of at least 10% tinnitus is associated with sleep difficulties, anxiety, depression, and reduced quality of life. The heterogeneity of tinnitus contributes to variability in treatment outcomes. Although numerous tinnitus interventions exist, it is not always clear which may be most suitable for the individual. Adapting to tinnitus may also change over time with some people habituating, some learning and using active coping strategies and others using passive coping styles such as withdrawal [9]. Identifying where a patient is on this continuum is important in order to adapt the intervention provided accordingly. It has also more recently been highlighted that significant others of those with tinnitus may experience third party disability [10]. When this is the case, they may not be in a good position to support those with tinnitus. 

Considering the dynamic interactions between a person’s health condition, environmental factors and social interactions, it is possible that these factors can affect the experience of tinnitus. From the social representations work done on hearing loss, cultural norms have been found to influence the experiences of hearing loss and hearing aids, and may thus also affect tinnitus experiences. Due to complexities associated with tinnitus, considering it from the SRT may be of value. In a recent study, we examined the social representations of ‘tinnitus’ and ‘health’ among individuals seeking online psychological interventions [11]. The most commonly occurring categories for tinnitus included descriptions of tinnitus (18%), annoying (13.5%), persistent (8%), and distracting (5%). The most commonly occurring categories for health included content (12%), conditions (8%), active (7%), take control (6%), and overweight (5%). The responses to tinnitus had predominantly negative connotations (i.e., 76.9%) whereas a larger proportion of responses towards their health was related to positive connotations (i.e., 46.4%). However, these analyses were conducted by looking at the responses of all the participants together. In previous studies on social representation of hearing loss and hearing aids, examining the patterns in responses using cluster analysis provided some additional insights that were not evident in a single group analysis [6,12]. In the current study, we therefore extended analysis of the social representation data from a previous study [11] to examine the patterns in social representation of tinnitus and its relation to demographics and clinical variables.

## 2. Method

### 2.1. Study Design

A cross sectional design was used. Ethical approval was obtained from the Institutional Review Board at Lamar University, Beaumont, TX, USA (IRB-FY17-209 and IRB-FY20-200-1).

### 2.2. Participants

Participants included were those with bothersome tinnitus seeking internet-based treatment for their tinnitus by signing up for treatment studies. This study was nested in three separate clinical trials of Internet-based cognitive behavioral therapy (ICBT) for tinnitus [13,14,15].

### 2.3. Data Collection

A series of structured as well as open-ended questionnaires were collected online via the ePlatform used for the internet intervention. The structured outcome measures included the Tinnitus Functional Index (TFI) as a measure of tinnitus distress, Generalized Anxiety Disorder—7 (GAD7 [16]) as a measure of anxiety, Patient Health Questionnaire-9 (PHQ9 [17]) as a measure of depression, Insomnia Severity Index (ISI) as a measure of insomnia, and the EQ-5D-5L Visual Analogue Scale (VAS) as a measure of health-related quality of life.

The social representation data were gathered using the free association task as a suggested method to access semantic content of social representation and used in several previous hearing-related studies [5,6,7,11,12,18,19]. Participants were asked to think of five words or phrases that came to their mind spontaneously when they thought about the stimulus ‘tinnitus’ and type them in the order of importance (i.e., most important word/phrase in the beginning). The method is known as a free association task and due to the spontaneous way in which responses are elicited, they are considered less controlled; hence, they provide a better understanding of what constitutes the semantic universe of the terms associated with tinnitus avoiding social desirability bias. 

### 2.4. Data Analysis

The data were analyzed using both qualitative (content analysis) and quantitative (i.e., Chi square analysis and cluster analysis) analyses which are commonly used in previous social representation studies in the area of health and disability [6,12]. The quantitative analyses were conducted using the text analysis software, IraMuTeQ (http://www.iramuteq.org/git/, accessed on 1 January 2020).

### 2.5. Content Analysis

The qualitative content analysis as described by Graneheim and Lundman (2004) [20] was used to analyze the response to free-association task. This involved the grouping of similar words (e.g., annoying, irritation, and nuisance) into a category (e.g., annoying). This was done independently by two researchers and cross-checked by one researcher. When tinnitus was described (e.g., ringing, loud, etc), these were classed into a category called ‘descriptions of tinnitus’. These results have been reported in our previous manuscript [9], although a summary is provided here to set a context for the additional quantitative analysis as outlined below. 

### 2.6. Cluster Analysis

A cluster analysis was done using the Reinert algorithm [21,22,23]. This method is a hierarchical divisive clustering. A correspondence analysis (equivalent to that of principal component analysis for categorical data) is made on the matrix and coordinates individuals on the first factor and uses the split matrix in two different groups (0/1 matrix). The produced clusters are groups of individuals who tend to be homogeneous in their answers to the free association task, and heterogeneous between clusters. In these steps, only categories of free associations are used. The secondary variables (demographic details) are crossed with the clusters a posteriori. This method has been described in detail in our previous manuscripts [6,12].

### 2.7. Chi-square Analysis

A chi-square analysis was carried out to explore whether there were any significant associations between the clusters identified and the clinical and demographic variables. The proportion of individuals with a particular category within the cluster is compared to the proportion of individuals with the category in the rest of the sample to determine which clinical and/or demographic variables are over-represented in each cluster. 

## 3. Results

### 3.1. Participants

There were 399 respondents of which 47% were below 56 years and 53% were above 56 years of age. In the sample 52% were female and 48% were male with the majority (60%) having a university degree as seen in Table 1. The majority had severe or significant tinnitus, and no or mild–moderate anxiety or depression.

### 3.2. Content Analysis

The key 39 categories emerged from content analysis of responses to the free association task about tinnitus (see Table 2; [11]). The most common categories pre-intervention were descriptions of tinnitus (18%), annoying (13.5%), persistent (8%), distracting (5%), frustrating (4.7%), and distressing (4.4%).

### 3.3. Cluster Analysis

The cluster analysis included 398 of the 399 respondents (i.e., 99.7%) as one participant was excluded as being identified as an outlier by the analysis software [6]. Four clusters were identified by the software. These clusters represented levels of the tinnitus experience in four stages, ranging from debilitating to acceptance of tinnitus (see Table 3). Cluster 1 represents those with the most debilitating tinnitus. Cluster 2 were the distressed group who were out of the debilitating stage and more hopeful of improvements. Cluster 3 were the annoyed tinnitus group who were frustrated by the tinnitus but the tinnitus was less intruding. Cluster 4 were the group who were becoming more acceptant of their tinnitus although it still caused difficulties. Each cluster is discussed below in turn.

#### 3.3.1. Cluster 1: Debilitating Tinnitus

A total of 97 of the 398 respondents (i.e., 24.4%) belonged to cluster 1, which was characterized mainly by debilitating. The responses more likely associated with this cluster included angering, miserable, isolating, debilitating, distressing, horrible, limiting, disturbing, depression, exhausting, and challenging. Chi square analysis suggested that participants who were Hispanic or Latino (*X^2^* = 18.8; *p* < 0.0001), had severe tinnitus (*X^2^* = 16; *p* < 0.0001), quality of life VAS scores below 80 (*X^2^* = 7.5; *p* = 0.0006), severe anxiety (*X^2^* = 7.3; *p* = 0.0006), entry level or unskilled work (*X^2^* = 6.3; *p* = 0.01), not working (*X^2^* = 5; *p* = 0.02), and high school level education (*X^2^* = 5.3; *p* = 0.02) were significantly more likely to be in this cluster.

#### 3.3.2. Cluster 2: Distressing Tinnitus

A total of 41 of the 398 respondents (i.e., 10.3%) belonged to cluster 2, which was characterized mainly by distressing tinnitus. The categories more likely to be responded by respondents from this cluster included hopeful, determination, tormenting, hopelessness, calming, unfair, frightening, horrible, loss of quiet, and depression. This category included some positive tinnitus representations not seen in cluster 1, e.g., hopeful, calming. Chi square analysis suggested that participants who were mixed race (*X^2^* = 6.58; *p* = 0.01), had moderate to severe insomnia (*X^2^* = 4.91; *p* = 0.03) and severe tinnitus distress (*X^2^* = 4.01; *p* = 0.045) were significantly more likely to be in this cluster.

#### 3.3.3. Cluster 3: Annoying Tinnitus

A total of 183 of the 398 respondents (i.e., 46%) belonged to cluster 3, which was characterized mainly by annoying tinnitus that was not at the previous levels of debilitating and distressing tinnitus. The categories more likely to be responded by respondents from this cluster included annoying, persistent, description of tinnitus, distracting, frustrating, healthy, need to sleep, interfering, and unintruding (see Table 3). As with cluster 2, there was a positive representation such as healthy. Chi square analysis suggested that participants from who were not Hispanic or Latino (*X^2^* = 17.3; *p* < 0.0001), quality of life VAS scores of 80 or above (*X^2^* = 10.4; *p* = 0.001), no anxiety (*X^2^* = 9.3; *p* = 0.002), significant tinnitus problem (*X^2^* = 7.6; *p* = 0.005), and no clinically significant insomnia (*X^2^* = 5.2; *p* = 0.02) were significantly more likely to be in this cluster.

#### 3.3.4. Cluster 4: Accepting Tinnitus

A total of 77 of the 398 respondents (i.e., 19.4%) belonged to cluster 4, which was characterized mainly by accepting tinnitus. The categories more likely to be responded to by respondents in this cluster included hearing difficulties, natural process, regret, helped, insomnia, treatment, description of tinnitus, and loss of quiet. The most positive associations are seen here with representations such as natural process, helped, and treatment. Chi square analysis suggested that participants of male gender (*X^2^* = 8.2; *p* = 0.0004) and being 56 years and older (*X^2^* = 4.3; *p* = 0.04) were significantly more likely to be in this cluster.

### 3.4. Tinnitus Intervention Suggestions

Interventions targeting individuals with tinnitus should consider their level of distress as well as associated difficulties. In this context, the characteristics of each cluster as well as the intervention suggestions for each of the groups are summarized in Table 4. 

## 4. Discussion

This study is one of the first to explore the patterns in the social representation of tinnitus reported by adults seeking help for bothersome tinnitus via an online intervention for tinnitus. A free association task resulted in 39 categories of responses such as tinnitus being annoying, persistent, and distracting. Four distinct clusters were identified, differing in the level of distress regarding tinnitus. These ranged from debilitating, distressing, and annoying to accepting tinnitus. The most frequent variables (responses) regarding tinnitus representations were annoying, persistent, and descriptions of tinnitus.

Each group was associated with unique demographic and clinical characteristics. Ethnicity was a factor for the three clusters most distressed by tinnitus. It could be that cultural barriers exist, such as different views on tinnitus that contribute to this finding. Being Hispanic or Latino was associated with being more likely to be in the debilitating or distressing tinnitus clusters. It may be that these ethnic groups have specific barriers to help-seeking which may exacerbate their difficulties. These may include the possibility of being economically disadvantaged and thus healthcare is not always accessible. These socio-demographic factors have been shown to impact on health-seeking in individuals with depression [24], and social and cultural expectations and norms were significant barriers to help-seeking behaviors for young adults. These possible cultural barriers should be considered during intervention planning as shown in the current study as well as highlighted by other researchers [25].

Clinical and research specialists coming across individuals with tinnitus should consider these categories and their associations. Listening to the language used to describe tinnitus may be insightful as to the level of tinnitus distress being experienced. The use of descriptions such as debilitating, isolating, angering, and being miserable should be taken seriously and timely interventions should be considered together with use of outcome measures to establish the level of distress. In this study around a quarter of the sample had debilitating tinnitus (i.e., cluster 1). The unique demographic characteristics of this group were those that were mostly educated to high school level and were not working or had entry level or unskilled level of work. These factors may make it more difficult to access healthcare information and help. Access may partly be dependent on levels of readability which are often high for health as well as tinnitus information [26,27]. Moreover, those not working or with less well paid jobs may not have the financial means to obtain help. This could exacerbate both the tinnitus and associated difficulties as indicated with this cluster having severe tinnitus and significant anxiety and lower levels of health-related quality of life (i.e., EQ-5D-5L VAS scores below 80). Due to the level of distress additional referrals should be considered to address comorbidities such as anxiety. Interventions should be immediate and supported by a professional. Psychological approaches such as cognitive behavioral therapy (CBT) which has the highest research evidence for tinnitus [28] are most likely important for this group due to the associated comorbidities.

It was interesting that the smallest cluster were those who were distressed by tinnitus and represented 10% of the sample (i.e., cluster 2). Their representation of tinnitus was that it was tormenting, frightening, horrible, unfair, helpless, etc. Unlike those in the debilitating tinnitus cluster, there were some more hopeful representations as well such as hopeful, determination, and calming. Being Hispanic or Latino was associated with being more likely to be in the debilitating or distressing tinnitus clusters and for this cluster also being of mixed race. This finding again highlights the possibility that these ethnic groups have specific barriers to help-seeking which may exacerbate their difficulties. The unique clinical characteristic of this group was moderate or severe insomnia. Poor sleep is often associated with tinnitus and can make tinnitus harder to manage and hence less bearable [29]. Hence, psychological approaches such as CBT for tinnitus are worth considering for this group due to the likely tinnitus comorbidities [30].

Most of the sample (46%) of those seeking help for their tinnitus were annoyed by their tinnitus (cluster 3). Their representation of tinnitus was that it was annoying, persistent, frustrating, and interfering. It is notable that in this cluster, there were more positive associations with tinnitus as well such as it being unintruding, need to stop, and healthy, again indicating some positive representation and hope. This group were also more likely to rather describe the tinnitus, e.g., a ringing. Those in this cluster were more likely not to be Hispanic nor Latino.

The tinnitus distress levels were also lower (e.g., significant) instead of severe. Their levels of insomnia were likely to be non-significant and they had higher levels of quality of life (i.e., EQ-5D-5L VAS above 80). Although they are more likely to have significant tinnitus, it did not result in anxiety or insomnia. This may be a good stage to start providing structured interventions to ensure the distress will not progress to distressed and/or debilitating levels. 

The final cluster represented those who were more accepting of their tinnitus, representing 20% of the sample (i.e., cluster 4). Their representation of tinnitus was describing the tinnitus, loss of quiet, hearing difficulties, regret, insomnia, and also positive representations such as treatment, helped, natural process. Their unique demographic characteristics were that they were more likely to be male and aged 56 years or older. There were no clinical variables associated with this cluster. This group still has difficulties although they appear to be in a stage of acceptance and possibly engagement with tinnitus treatments due to the representation of treatment and help that were found. Alternatively, individuals in this group may also be at an early stage of their condition and could benefit from informational counseling.

According to the SRT, the way we perceive our surrounding world and our knowledge affects how we act. It is important to identify the social concepts that are related to a particular object and separate the factors that impede the preferred practice by a way of intervention [31]. Media such as campaigns play an important role in developing new knowledge about a particular topic and this will help to amend or create new social representations [32]. Personal beliefs about tinnitus are going to shape how people respond to their symptoms. Furthermore, societal factors such as societal attitudes, norms and practices are reported to influence the help-seeking and rehabilitation uptake in people. Hence, our efforts to support those with tinnitus needs to address not only the individual but also psychological and social factors that could be contributing. This includes significant others in their lives and societal attitudes. Enabling significant others to understand tinnitus and support those with tinnitus and also manage third-party disability may indirectly help those with tinnitus [10]. Public health campaigns prioritizing awareness of tinnitus and provision of more resources should be aimed for together with interventions. Such campaigns should be established especially in communities where help seeking may be difficult. 

An unanswered question within tinnitus research is that of why some people effectively habituate and are not bothered by their tinnitus and others do not. It may be that these social values and social support contribute to this and more work considering the importance of social support in tacking tinnitus is needed. 

This work also has a wider implication that social representation can also be used for tinnitus prevention campaigns. Social representation of loud music has indicated that loud music is perceived to have both positive and negative aspects within society and culture, and the harmful effects to hearing healthcare were not always present. [18] Work such as this can be used by prevention campaigns to educate and change societal norms.

What is interesting is that the free association task identified very different social representations than those identified in the context of hearing loss and hearing aids [5,6,7] management, causes of hearing loss, communication difficulties, disability, hearing ability or disability, hearing instruments, negative mental state, the attitudes of others, and sound and acoustics of the environment. The representations for hearing aids were cosmetic aging and disability, appearance and design, cost, hearing instruments and improved hearing and communication. This may indicate the clear pathways that are generally available to assess, manage, and provide hearing instruments for hearing loss. The social representations were generally external for hearing loss which is in direct contrast to the majority of the social representations for tinnitus being internal, e.g., angering, miserable, isolating, and debilitating. These internal factors are often more difficult to manage than external factors, and may contribute to the distress experienced by those with tinnitus. Due to the high prevalence of tinnitus, and lack of expertise in many geographical locations to appropriately manage tinnitus, many with tinnitus may be unable to access the needed care, which may exacerbate the difficulties associated with tinnitus. 

### Study Limitations, Clinical Implications, and Future Directions

This sample represents those that were seeking help for their tinnitus and thus does not represent the general tinnitus population who include many people who are not distressed by their tinnitus. There may have been a sampling bias in the study sample. Moreover, there may be demographic and clinical variables that were not explored that may contribute to these clusters. This includes the presence of hearing loss which was not objectively assessed due to the study being online. Future studies should assess the association between each Cluster and the degree of hearing loss to identify any patterns. Chi-square analysis was used to explore the association between each cluster and the demographic and clinical variables. In future studies further associations such as hearing ability can be approached using multivariate analysis methods.

This study indicates the importance of providing individuals bothered by tinnitus time to discuss their tinnitus and carefully listen to the language used to describe their tinnitus. These descriptions together with other assessment means such as the use of self-reported outcome measures may help triage these individuals to the most appropriate care. Further implications are that social norms and values may partly determine the individual’s responses to difficulties such as tinnitus and how help-seeking is approached. Health care services should be mindful that help seeking may be more difficult in certain social settings and work on addressing such barriers is particularly important for these groups. Moreover, to address tinnitus, the focus needs to be on not only helping the individual, but also helping those closest to them. Enabling wider societies to understand the impact of tinnitus can assist them in responding to tinnitus difficulties in a more timely and appropriate manner. Future work should be directed to identifying whether social representation of tinnitus changes following undertaking a tinnitus intervention. It may be that these social representations depict the journey from debilitating tinnitus to being able to accept the tinnitus.

## Figures and Tables

**Table 1 brainsci-12-01221-t001:** Demographic details (*n* = 399).

Variable	*n* (%)
Age (in years)	
Below 56 years	187 (46.9%)
56 years and above	212 (53.1%)
Gender	
Female	208 (52.1%)
Male	191 (47.9%)
Ethnicicy	
Hispanic or Latino	42 (11%)
Not Hispanic or Latino	357(89%)
Education	
Less than high school	5 (1.2%)
High school	41 (10.3%)
Some college but not a degree	115 (28.8%)
A university degree	238 (59.7%)
Work status	
Entry level or unskilled	13 (3.3%)
Skilled or professional work	242 (60.6%)
Retired	112 (28.1%)
Not working	32 (8%)
Tinnitus duration	
Less than 1 year	44 (11%)
1–5 years	126 (31.6%)
More than 5 years	229 (57.4%)
Tinnitus severity (TFI)	
Mild tinnitus	47 (11.8%)
Severe tinnitus	213 (53.4%)
Significant problem	139 (34.8%)
Anxiety (GAD-7)	
No anxiety	177 (44.4%)
Mild to moderate anxiety	178 (44.6%)
Severe anxiety	44 (11.03)
Depression (PHQ-9)	
No depression	163 (40.8%)
Mild to moderate depression	175 (43.9%)
Moderately severe to severe depression	61 (15.3%)
Insomnia (ISI)	
No clinically significant insomnia	128 (32.1%)
Subthreshold insomnia	124 (31.1%)
Moderate to severe clinical insomnia	147 (36.8%)
Quality of life (EQ-5D-5L VAS)	
VAS score below 80	198 (49.6%)
VAS scores of 80 and above	201 (50.4%)

**Table 2 brainsci-12-01221-t002:** Key categories identified by content analysis in alphabetical order.

Category (Examples)	Number and Frequency of Occurrence	
	*n*	%
Accepting (e.g., just deal with it, it is what it is, don’t think much)	45	2.37
Angering (e.g., anger, rage, hate)	46	2.42
Annoying (e.g., annoying, irritation, nuisance)	257	13.53
Bothersome	52	2.74
Calming	7	0.37
Challenging	28	1.47
Debilitating	45	2.37
Depressing	29	1.53
Description of tinnitus	342	18
Determination	14	0.74
Distracting	96	5.05
Distressing	83	4.37
Disturbing	5	0.26
Exhausting	35	1.84
Frightening	36	1.89
Frustrating	68	3.58
Healthy	4	0.21
Hearing difficulties	73	3.84
Helped	6	0.32
Helplessness	48	2.53
Hopeful	28	1.47
Horrible	56	2.95
Insomnia	21	1.11
Interesting	4	0.21
Interfering	32	1.68
Isolating	40	2.11
Limiting	10	0.53
Loss of quiet	37	1.95
Miserable	45	2.37
Natural process	21	1.11
Need to stop	43	2.26
Persistent	151	7.95
Regret	12	0.63
Tormenting	27	1.42
Treatment	5	0.26
Unbearable	8	0.42
Uncontrollable	15	0.79
Unfair	18	0.95
Unintruding	8	0.42

**Table 3 brainsci-12-01221-t003:** Clusters based on responses to free association task.

Cluster 1 (24.4%): Debilitating Tinnitus (Level 1 Distress)	Cluster 2 (10.3%): Distressing Tinnitus (Level 2 Distress)
Angering	Hopeful
Miserable	Determination
Isolating	Tormenting
Debilitating	Hopelessness
Distressing	Calming
Horrible	Unfair
Limiting	Frightening
Disturbing	Horrible
Depression	Loss of quiet
Exhausting	Depression
Challenging	
**Cluster 3 (46%): Annoying Tinnitus (Level 3 Distress)**	**Cluster 4 (19.4%): Accepting Tinnitus (Level 4 Distress)**
Annoying	Hearing difficulties
Persistent	Natural process
Description of tinnitus	Regret
Distracting	Helped
Frustrating	Insomnia
Healthy	Treatment
Need to stop	Description of tinnitus
Interfering	Loss of quiet
Unintruding	

**Table 4 brainsci-12-01221-t004:** Summary of the characteristics for each subgroup and suggested intervention guidelines based on these subgroups.

	Cluster 1 (24.4%): Debilitating Tinnitus (Level 1 Distress)	Cluster 2 (10.3%): Distressing Tinnitus (Level 2 Distress)	Cluster 3 (46%): Annoying Tinnitus (Level 3 Distress)	Cluster 4 (19.4%): Accepting Tinnitus (Level 4 Distress)
Representation	Angering, miserable, isolating, debilitating	Hopeful, determination, tormenting, hopelessness	Annoying, persistent, descriptions of tinnitus, Distressing	Hearing difficulties, natural process, regret, helped
**Demographic and clinical variables**				
Ethnicity	Being Hispanic or Latino	Mixed race or Hispanic or Latino	NOT Hispanic or Latino	
Gender and age				Male and Older than 56 years
Education:	High school level education			
Work status:	Entry level or unskilled work			
Tinnitus	Severe tinnitus (TFI > 50/100)	Severe tinnitus (TFI > 50/100)	Significant tinnitus (TFI: 25–50/100)	
Anxiety	Severe		No anxiety	
Insomnia		Moderate or severe	Not significant	
Quality of Life VAS Score:	Lower (below 80)		Higher (above 80)	
**Tinnitus intervention suggestions**				
Intensity and support	Immediate and intense support. Support group possibly with similar demographic characteristics could be helpful	Immediate and intense support. Support group possibly with similar demographic characteristics could be helpful	Timely support due to significant levels of tinnitus	Less intensive intervention
Intervention format	Immediate, regular close guidance by a professional	Immediate, regular guidance by a professional	Regular guidance provided by a professional	Consider self-help or self-guided if suitable and appropriate
Referrals	Consider psychological support for additional mental health concerns such as anxiety.	Consider additional support for insomnia		
Monitoring	Close monitoring of anxiety, depression and tinnitus distress initially on a weekly basis	Close monitoring of insomnia anxiety, depression and tinnitus distress initially on a weekly basis	Monitoring using independent self-reported means	Monitoring using independent self-reported means

## Data Availability

The data presented in this study are available on request from the corresponding author. The data are not publicly available due to privacy concerns.

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
