# Peer review of "Experiential Characteristics among Individuals with Tinnitus Seeking Online Psychological Interventions: A Cluster Analysis"

_brainsci, 2022, doi:10.3390/brainsci12091221_

Round 1

Reviewer 1 Report

This study investigated the associations between social representations of tinnitus and demographic and clinical factors using a cluster analysis. It is interesting because tinnitus was analyzed with a new perspective in this study. However, there are limitations in this study.

1. Although this study was performed in the context of social representations, hearing thresholds for all the participants with tinnitus should be evaluated and presented in Table 1 because hearing loss is commonly associated with tinnitus. If the specific values of hearing thresholds could not be presented, the degree of hearing loss (e.g. normal, mild, moderate, severe hearing loss, etc.) should be presented.

2. In addition, the association between each Cluster and the degree of hearing loss should be analyzed and the significant values should be presented in Table 4.

3. In line 177, ethnicity distribution should be presented in Table 1 because ethnicity (Hispanic or Latino) was presented as significant variables in Table 4.

4. This study has a limitation because the demographic and clinical variables were not analyzed with multivariate methods. These variables could be related among them. Thus, the significance of these values might be different in multivariate analyses. If multivariate analyses could not be performed, it should be described as a limitation in Discussion section.

5. In line 113, was obtained was obtained -> was obtained.

6. In line 119, explain the abbreviation of ICBT.

7. In line 143, present the names of company, city, country of IraMuTeQ.

8. In line 177, Table 1, explain the abbreviation of CAS.

9. In line 296, it seems that ‘not to be not Hispanic or Latino’ should be modified to ‘not to be Hispanic nor Latino.’

Author Response

Many thanks for the time taken to review this paper and for the excellent suggestions. These have been addressed as indicated in the attached file.

Reviewer 2 Report

Beukes et al. performed a retrospective survey to identify and quantify the characteristics of subjects’ tinnitus. They used the same methodology, clustering of free-association task data, as previous publications by the same group, and identified 4 subgroups of tinnitus sufferers. Intervention strategies were suggested and discussed. The study is potentially helpful for guiding clinicians as well as providing epidemiological insight into tinnitus. However, the presentation is somewhat unclear. 

First of all, I do not think “social representation” is an accurate description of the study’s design/outcome. Looking at the semantic contents closely, one notices that none of the associated words reflect anything “social.” The cited studies from the same group that applied the method to studying hearing loss and hearing aid dealt with some aspects of cultural/social judgment in individuals (e.g., comparing context in different countries). The present study, however, did not set out to examine social context from the outset. In line 49, the author referred to “systems of values, ideas, and practices [that] influence worldviews.” I have trouble locating such an aspect herein. The word clusters express severity, life impact, and personal experience of tinnitus rather than values and worldviews. It can also be argued that a few of these descriptors are already encompassed by TFI. As such, I strongly suggest presenting the study as “experiential characteristics of tinnitus,” or “demographic”/“psychological” rather than “social.”

My second question is regarding the Reinert algorithm: why 4 clusters? Is it automatically detected, and if so, what criteria? If user-selected, why 4? Similarly, it is unclear why the age of 56 year old was used as a cutoff. I have trouble finding the rationale besides that it increases the separation of the 4 clusters. And if this is the case, can one identify more or fewer clusters using other arbitrary cutoffs?

I also find a huge oversight in the author’s discussion of ethnicity (i.e., the fact that Hispanic populations have more debilitating tinnitus). The authors were quick to evoke “social” and “cultural barriers” rather than pointing to the painfully obvious fact that these same populations are economically disadvantaged, a discussion that came much later about “job status” and “education level” and was treated in isolation from the subject’s ethnic background. I do not expect the authors to propose intervention strategies that actually involve social aspects, such as public health policies or advocating for access equity, but one cannot answer the question “why some people habituate to tinnitus and some do not” without, as the authors attempted but failed to grasp the essence of, a discussion on society.

Author Response

(The authors gave the same response as above.)
